# Comprehensive understanding of multiple resonance thermally activated delayed fluorescence through quantum chemistry calculations

Katsuyuki Shizu[1] & Hironori Kaji [1✉]

Molecules that exhibit multiple resonance (MR) type thermally activated delayed fluorescence (TADF) are highly efficient electroluminescent materials with narrow emission spectra. Despite their importance in various applications, the emission mechanism is still controversial. Here, a comprehensive understanding of the mechanism for a representative MR-TADF molecule (5,9-diphenyl-5,9-diaza-13b-boranaphtho[3,2,1-de]anthracene, DABNA-1) is presented. Using the equation-of-motion coupled-cluster singles and doubles method and Fermi's golden rule, we quantitatively reproduced all rate constants relevant to the emission mechanism; prompt and delayed fluorescence, internal conversion (IC), intersystem crossing, and reverse intersystem crossing (RISC). In addition, the photoluminescence quantum yield and its prompt and delayed contributions were quantified by calculating the population kinetics of excited states and the transient photoluminescence decay curve. The calculations also revealed that TADF occurred via a stepwise process of 1) thermally activated IC from the electronically excited lowest triplet state $T_1$ to the second-lowest triplet state $T_2$, 2) RISC from $T_2$ to the lowest excited singlet state $S_1$, and 3) fluorescence from $S_1$.

[1] Institute for Chemical Research, Kyoto University, Uji, Kyoto 611-0011, Japan. ✉email: kaji@scl.kyoto-u.ac.jp

Since highly efficient thermally activated delayed fluorescence (TADF) was used in organic light-emitting diodes (OLEDs)[1], many TADF molecules containing various electron donor and acceptor groups have been developed[2–9]. An important factor for TADF efficiency is the energy difference between the lowest triplet state ($T_1$) and the lowest excited singlet state ($S_1$), $\Delta E(T_1 \rightarrow S_1)$; a small $\Delta E(T_1 \rightarrow S_1)$ (<200 meV) is required to induce $T_1 \rightarrow S_1$ reverse intersystem crossing (RISC). Another important factor is a large transition-dipole moment between $S_1$ and ground-state $S_0$ to accelerate the rate of $S_1 \rightarrow S_0$ fluorescence. Experimentally, efficient RISC and high photoluminescence quantum yields (PLQYs) have been simultaneously realized by combining suitable donor and acceptor units that control the spatial overlap between the highest-occupied molecular orbitals (HOMOs) and the lowest-unoccupied molecular orbitals (LUMOs)[1–9]. It is important to note that the fluorescence spectra were broadened because of the charge-transfer character of $S_1$, which was a significant drawback for OLED applications in displays.

In 2016, Hatakeyama et al. developed a new class of TADF molecules[10]. Using a triphenylboron core possessing two nitrogen atoms (5,9-diphenyl-5,9-diaza-13b-boranaphtho[3,2,1-*de*]anthracene, DABNA-1, Fig. 1a), they observed a HOMO–LUMO separation without a conventional donor–acceptor structure, providing efficient TADF with a narrow emission spectrum. The HOMO–LUMO separation resulted from a multiple resonance (MR) effect, that is an opposite resonance effect induced by the boron and nitrogen atoms. MR–TADF molecules emitting blue-to-red fluorescence have been reported[11–39].

Several theoretical studies have attempted to reveal the TADF mechanism in MR molecules. Northey et al. investigated an intersystem-crossing (ISC) mechanism in DABNA-1 using quantum dynamics and time-dependent density-functional theory

(TD-DFT) at the PBE0/6-31 G(d) level[40]. There was only a 0.02-eV energy difference between $T_2$ and $S_1$, $\Delta E(T_2 \rightarrow S_1)$. However, $\Delta E(T_1 \rightarrow S_1)$ and $\Delta E(T_1 \rightarrow T_2)$ were large, 0.59 eV and 0.61 eV, respectively, which could not explain efficient RISC from $T_1$ to $S_1$. Gao et al. examined the density-functional dependence of $\Delta E(T_1 \rightarrow S_1)$ within the framework of TD-DFT[41]. The MPWK1CIS functional reproduced the experimental $\Delta E(T_1 \rightarrow S_1)$[41], as reviewed by Suresh et al.[21]. However, the RISC rate constant ($k_{RISC}$) was not calculated and the TADF mechanism was unclear. Also, they considered TADF in terms of direct (one-step) $T_1 \rightarrow S_1$ RISC, which differed from the work of Northey et al[40]. Pershin et al. reported that TD-DFT methods overestimated $\Delta E(T_1 \rightarrow S_1)$ for MR molecules, and that the spin-component-scaling second-order approximate coupled-cluster (SCS-CC2) method outperformed TD-DFT for predicting $\Delta E(T_1 \rightarrow S_1)$[42,43]. The partial inclusion of double excitations within the SCS-CC2 method was responsible for the improved accuracy in predicting $\Delta E(T_1 \rightarrow S_1)$. The SCS-CC2 calculation of DABNA-1-based molecules revealed the relationship between its molecular structure and electronic properties, $\Delta E(T_1 \rightarrow S_1)$, and the fluorescence rate constant. However, the TADF mechanism was still unclear because the $k_{RISC}$ was not calculated. Lin et al. calculated rate constants for fluorescence ($k_F$), ISC ($k_{ISC}$), $k_{RISC}$, and internal conversion (IC) ($k_{IC}$) for DABNA-1[44]. However, the calculated $k_{RISC}$ was $6.7 \times 10^2 \, \text{s}^{-1}$, which was much less than the experimental value of $1.0 \times 10^4 \, \text{s}^{-1}$[10]. The calculated $k_{ISC}$ of $1.4 \times 10^4 \, \text{s}^{-1}$ was two orders of magnitude less than the experimental value of $4.5 \times 10^6 \, \text{s}^{-1}$[10]. In addition, the calculated nonradiative decay, $k_{IC}(S_1 \rightarrow S_0)$, was greater than the $k_F(S_1 \rightarrow S_0)$, suggesting that the PLQY of DABNA-1 was less than 50%, in contrast with the 88% experimental value[10]. All these previous studies[40–44] only partially explained the photophysical properties of MR-TADF mainly because crucial RISC was not elucidated.

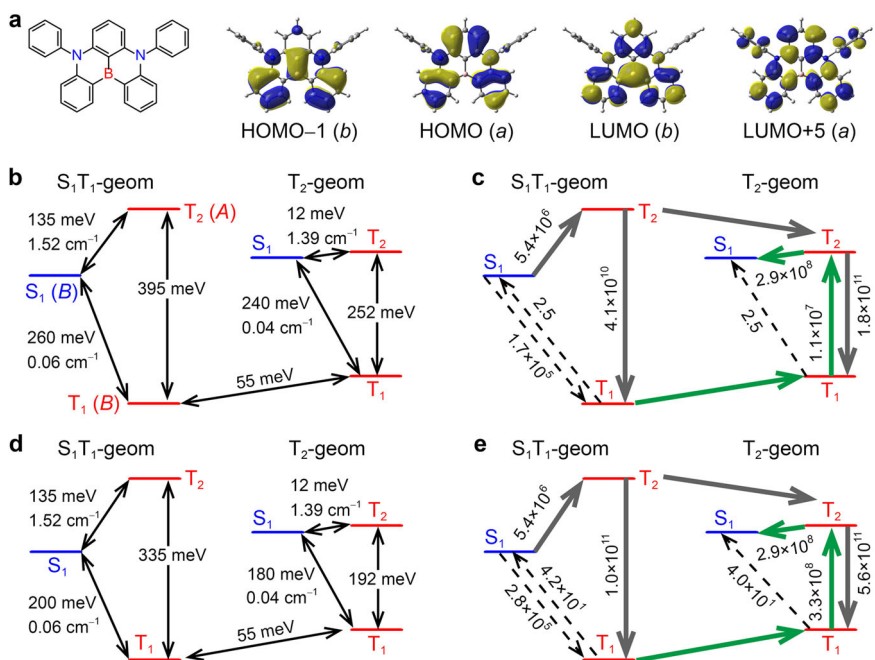

**Fig. 1 Molecular orbitals and electronic transitions of DABNA-1. a** Structure of DABNA-1 and HOMO-1, HOMO, LUMO, and LUMO+5 distributions calculated at the RHF/6-31 G level (HOMO = highest-occupied molecular orbital, LUMO = lowest-unoccupied molecular orbital). The orbital symmetry is shown in parentheses. **b** Energy differences (meV) between $S_1$, $T_1$, and $T_2$. $S_1$–$T_1$ and $S_1$–$T_2$ spin–orbit couplings (cm$^{-1}$). The state symmetry is shown in parentheses. **c** The stepwise $S_1 \rightarrow T_2 \rightarrow T_1$ and $T_1 \rightarrow T_2 \rightarrow S_1$ processes of DABNA-1. The gray solid arrow depicts stepwise $S_1 \rightarrow T_2 \rightarrow T_1$; the green solid arrow depicts stepwise $T_1 \rightarrow T_2 \rightarrow S_1$; the dotted arrows show minor decay and upconversion processes. The values are calculated rate constants for one-step transitions. **d** The energy differences calculated using the corrected $T_1$ energy. **e** Calculated rate constants for one-step transitions using the corrected $T_1$ energy.

Here, we report a comprehensive understanding of the TADF mechanism using the representative MR molecule, DABNA-1. So far, $\Delta E(T_1 \rightarrow S_1)$ (and spin–orbit coupling (SOC) in some cases) and oscillator strength have been considered to understand the TADF mechanism and to design TADF molecules. However, these are not sufficient for the above aim; quantifications of all types of rate constants and those of energy levels, including higher-lying states relevant to the emission process, are required for the complete understanding of the emission mechanism. Recently, we reported rate-constant predictions enabled by a proposed cost-effective calculation method based on Fermi's golden rule[45]. Using the method, we calculated $k_{IC}$, $k_{ISC}$, $k_{RISC}$, $k_F$, and the rate constant for phosphorescence ($k_{Phos}$), of benzophenone as an example. The calculated rate constants agreed well with experimental values. Here, we applied the same method to DABNA-1 and determined all relevant rate constants. We also determined the lifetime of the delayed component in transient PL (trPL), the rate constant of TADF ($k_{TADF}$), and the PLQY, by calculating the population kinetics of the excited states and the trPL decay curve. The calculations also indicated that, after photoexcitation, $T_1$ was first generated via $S_1 \rightarrow T_2$ ISC and $T_2 \rightarrow T_1$ IC. Then, triplet-to-singlet conversion occurred via thermally activated $T_1 \rightarrow T_2$ IC and $T_2 \rightarrow S_1$ RISC.

## Results and discussion

**Excited states of DABNA-1.** $S_0$, $S_1$, $T_1$, and $T_2$ geometries of DABNA-1 were obtained at the TPSSh/6-31+G(d) level. Then, excited states were computed using the equation-of-motion coupled-cluster singles and doubles (EOM-CCSD) method (see "Methods" section). At the TD-TPSSh level, $S_1$ and $T_1$ local-energy minima were located at the $S_1$- and $T_1$-optimized geometries, respectively. However, as shown in Supplementary Table 10–12 and Supplementary Fig. 3, the lowest $S_1$ and $T_1$ energy levels calculated at the EOM-CCSD level were both located at the $T_1$ geometry (3.37 and 3.12 eV, respectively). In addition, the $S_1$ and $T_1$ structures were nearly the same, including the dihedral angles (Supplementary Table 5), hence, the structure was denoted the $S_1T_1$ geometry. Because the Stokes shift was experimentally observed for DABNA-1 in solution and in solid films[10], it was reasonably assumed that fluorescence occurs in the $S_1$ structure at the adiabatic energy minimum. The $S_0$, $S_1$, $T_1$, and $T_2$ electronic states and the $S_1T_1$ and $T_2$ equilibrium geometries were used to model TADF for DABNA-1 (eight electronic states in all). Because $S_2$ states were 0.89 eV higher in energy than the $S_1$ states, their contributions were neglected (Supplementary Fig. 3). The $T_3$ states were only 0.2 eV higher than $S_1$, but were neglected because the electronic-transition rate constants from $S_1$ to $T_3$ were far smaller than those of the competing processes ($S_1$–$T_1$ and $S_1$–$T_2$) and that from $T_2$ to $T_3$ was smaller than that from $T_2$ to $T_1$ (Supplementary Fig. 3). In the EOM-CCSD calculations, $S_1$ and $T_1$ involved predominantly HOMO → LUMO transitions, whereas $T_2$ involved predominantly a linear combination of HOMO → LUMO + 5 and HOMO − 1 → LUMO transitions (Fig. 1a, Supplementary Fig. 2, and Supplementary Tables 6–9). Figure 1b shows a calculated energy-level diagram of DABNA-1. The adiabatic $T_1 \rightarrow S_1$ energy difference calculated with the EOM-CCSD/6-31 G method ($S_1T_1$ geometry) was only 0.06 eV larger than the experimentally obtained 0.2 eV. Thus, EOM-CCSD significantly improved the overestimation of $\Delta E(T_1 \rightarrow S_1)$, relative to those by previously reported TD-DFT methods[40,42–44].

## Calculation of rate constants and luminescence quantum efficiencies. 
We examined triplet formation from $S_1$. Figure 1b shows calculated energy differences and SOCs, and Table 1 lists calculated and experimental values of $k_{IC}$, $k_{ISC}$, $k_{RISC}$, $k_F$, and $k_{Phos}$. The

**Table 1 Calculated and experimental photophysical properties of DABNA-1.**

| | Calc. | | Expt. |
|---|---|---|---|
| | Raw* | Corrected** | |
| $\Delta E(T_1 \rightarrow S_1)$ | 240 | 180 | |
| $\Delta E(T_2 \rightarrow S_1)$ | −12 | −12 | |
| $\Delta E(T_1 \rightarrow T_2)$ | 252 | 192 | 196 |
| $k_F(S_1 \rightarrow S_0)$ | $1.4 \times 10^8$ s$^{-1}$ | $1.4 \times 10^8$ s$^{-1}$ | $1.0 \times 10^8$ s$^{-1}$ |
| $k_{IC}(S_1 \rightarrow S_0)$ | $1.2 \times 10^7$ s$^{-1}$ | $1.2 \times 10^7$ s$^{-1}$ | $1.3 \times 10^7$ s$^{-1}$ |
| $k_{ISC}(S_1 \rightarrow T_1)$ | $1.7 \times 10^5$ s$^{-1}$ | $2.8 \times 10^5$ s$^{-1}$ | |
| $k_{ISC}(S_1 \rightarrow T_2)$ | $5.4 \times 10^6$ s$^{-1}$ | $5.4 \times 10^6$ s$^{-1}$ | |
| $k_{IC}(T_2 \rightarrow T_1)$ | $1.8 \times 10^{11}$ s$^{-1}$ | $5.6 \times 10^{11}$ s$^{-1}$ | |
| $k_{RISC}(T_2 \rightarrow S_1)$ | $2.9 \times 10^8$ s$^{-1}$ | $2.9 \times 10^8$ s$^{-1}$ | |
| $k_{Phos}(T_1 \rightarrow S_0)$ | $9.4 \times 10^{-1}$ s$^{-1}$ | $1.9 \times 10^{-1}$ s$^{-1}$ | |
| $k_{IC}(T_1 \rightarrow T_2)$ | $1.1 \times 10^7$ s$^{-1}$ | $3.3 \times 10^8$ s$^{-1}$ | |
| $k_{RISC}(T_1 \rightarrow S_1)$ | 2.5 s$^{-1}$ | $4.2 \times 10^1$ s$^{-1}$ | |
| $\Phi$ | 0.92 | 0.92 | 0.88 |
| $\Phi_p$ | 0.89 | 0.89 | 0.85 |
| $\Phi_{TADF}$ | $2.9 \times 10^{-2}$ | $3.2 \times 10^{-2}$ | $3.5 \times 10^{-2}$ |
| $\Phi_{ISC}$ | $3.5 \times 10^{-2}$ | $3.5 \times 10^{-2}$ | $4.0 \times 10^{-2}$ |
| $\tau_{TADF}$ | $5.2 \times 10^{-4}$ s | $5.1 \times 10^{-5}$ s | $9.4 \times 10^{-5}$ s |
| $k_{TADF}$ | $0.16 \times 10^4$ s$^{-1}$ | $1.8 \times 10^4$ s$^{-1}$ | $0.94 \times 10^4$ s$^{-1}$ |
| $k(T_1 \rightarrow T_2 \rightarrow S_1)$ | $0.17 \times 10^4$ s$^{-1}$ | $1.9 \times 10^4$ s$^{-1}$ | $1.0 \times 10^4$ s$^{-1}$ |
| $k(S_1 \rightarrow T_2 \rightarrow T_1)$ | $5.4 \times 10^6$ s$^{-1}$ | $5.4 \times 10^6$ s$^{-1}$ | $4.5 \times 10^6$ s$^{-1}$ |

Experimental values are from Hatakeyama et al.[10].
*Energy differences were calculated using the EOM-CCSD/6-31 G method (Fig. 1b,c).
**Corrected energy differences were used (Fig. 1d,e).

raw rate constants in Table 1 were calculated using excitation energies calculated via EOM-CCSD. Corrected values are discussed below. Equations for $k_{IC}$, $k_{ISC}$, $k_{RISC}$, $k_F$, and $k_{Phos}$ are given in Supplementary Equations S1–S9 and Supplementary Fig. 4.

Calculated values of $k_F(S_1 \rightarrow S_0)$ ($1.4 \times 10^8$ s$^{-1}$) and $k_{IC}(S_1 \rightarrow S_0)$ ($1.2 \times 10^7$ s$^{-1}$) agreed quantitatively with experimental results ($1.0 \times 10^8$ s$^{-1}$ and $1.3 \times 10^7$ s$^{-1}$, respectively) determined by Hatakeyama et al.[10] from PL decay curves of a 1-wt% DABNA-1: 9,9′-biphenyl-3,3′-diylbis-9H-carbazole film at 300 K (Table 1). The experimentally obtained $k_{ISC}$ value ($4.5 \times 10^6$ s$^{-1}$) agreed with $k_{ISC}(S_1 \rightarrow T_2)$ ($5.4 \times 10^6$ s$^{-1}$), rather than $k_{ISC}(S_1 \rightarrow T_1)$ ($1.7 \times 10^5$ s$^{-1}$), suggesting that the experimental $k_{ISC}$ should be assigned to $k_{ISC}(S_1 \rightarrow T_2)$. The $k_{ISC}(S_1 \rightarrow T_2)$ was ten times greater than $k_{ISC}(S_1 \rightarrow T_1)$, because of the larger $S_1$–$T_2$ SOC ($1.52$ cm$^{-1}$) relative to that of $S_1$–$T_1$ ($0.06$ cm$^{-1}$). The large $S_1 \rightarrow T_2$ SOC enhanced $S_1 \rightarrow T_2$ ISC, despite the uphill transition from $S_1$ to $T_2$ (the Franck–Condon energy difference was 135 meV, Fig. 1b and Supplementary Table 13), compared with the downhill transition from $S_1$ to $T_1$ (260 meV). The small $S_1$–$T_1$ SOC resulted from very similar $S_1$ and $T_1$ orbital configurations (HOMO → LUMO transition). These results suggested that the ISC ($S_1 \rightarrow T_1$ conversion) of DABNA-1 occurred via the stepwise $S_1 \rightarrow T_2 \rightarrow T_1$ process, rather than by the one-step $S_1 \rightarrow T_1$ ISC, irrespective of the geometry relaxation (gray arrows in Fig. 1c). Such $T_2$-supported ISC is experimentally observed for a TADF system when $\Delta E(S_1 \rightarrow T_2)$ is sufficiently small[46]. Because $T_2 \rightarrow T_1$ IC was much faster than $S_1 \rightarrow T_2$ ISC, the latter was the rate-determining step of the $S_1 \rightarrow T_2 \rightarrow T_1$ process (Fig. 1c). Hence, the net rate constant $k(S_1 \rightarrow T_2 \rightarrow T_1)$ was approximately equal to $k_{ISC}$ ($S_1 \rightarrow T_2$) = $5.4 \times 10^6$ s$^{-1}$ (Table 1).

For RISC from $T_1$, the calculated $k_{RISC}(T_1 \rightarrow S_1)$ ($2.5$ s$^{-1}$) was considerably less than the experimental value ($1 \times 10^4$ s$^{-1}$)[10], indicating that $T_1 \rightarrow S_1$ conversion did not occur in one-step but via a more effective process. Because the $S_2$ and $T_3$ energy levels, as well as higher singlet and triplet states, could not contribute to the RISC from $T_1$, as discussed above, and $T_2$ was close to $S_1$ (see

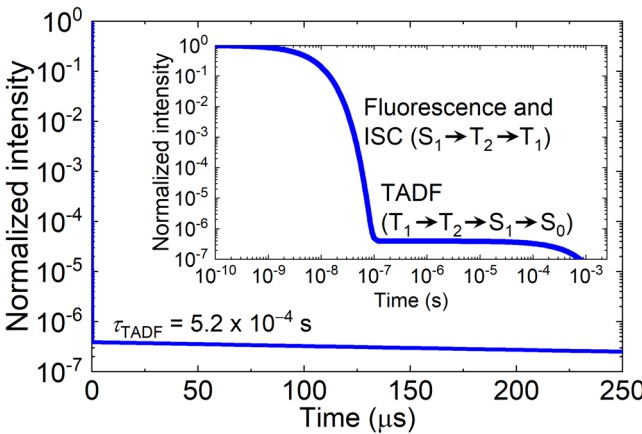

**Fig. 2 Calculated transient photoluminescence (trPL) decay curve.** The inset is a log–log plot of the decay curve.

Supplementary Fig. 3), stepwise $T_1 \rightarrow T_2 \rightarrow S_1$ was the most promising mechanism (green arrows in Fig. 1c). To determine $k(T_1 \rightarrow T_2 \rightarrow S_1)$, we calculated the trPL decay curve (Fig. 2) via kinetic equations (Supplementary Equations S10–S17, Supplementary Table 15, and Supplementary Note 1). The total PLQY ($\Phi$) and the TADF lifetime ($\tau_{TADF}$) were then obtained from the trPL decay curve (Fig. 2). The prompt and delayed components of $\Phi$ ($\Phi_P$ and $\Phi_{TADF}$, respectively) were calculated from

$$\Phi_P = \frac{k_F(S_1 \rightarrow S_0)}{k_F(S_1 \rightarrow S_0) + k_{IC}(S_1 \rightarrow S_0) + k_{ISC}(S_1 \rightarrow T_2) + k_{ISC}(S_1 \rightarrow T_1)} \quad (1)$$

$$\Phi_{TADF} = \Phi - \Phi_P \quad (2)$$

and the ISC quantum yield ($\Phi_{ISC}$) was calculated from

$$\Phi_{ISC} = \frac{k_{ISC}(S_1 \rightarrow T_2) + k_{ISC}(S_1 \rightarrow T_1)}{k_F(S_1 \rightarrow S_0) + k_{IC}(S_1 \rightarrow S_0) + k_{ISC}(S_1 \rightarrow T_2) + k_{ISC}(S_1 \rightarrow T_1)} \quad (3)$$

Then, $k_{TADF}$ was determined from[10]

$$k_{TADF} = \frac{\Phi_{TADF}}{\Phi_{ISC} \tau_{TADF}} \quad (4)$$

Finally, $k(T_1 \rightarrow T_2 \rightarrow S_1)$ was calculated from

$$k(T_1 \rightarrow T_2 \rightarrow S_1) = \frac{k_F(S_1 \rightarrow S_0) \times k_{TADF}}{k_F(S_1 \rightarrow S_0) - k(S_1 \rightarrow T_2 \rightarrow T_1)} \quad (5)$$

Table 1 also lists calculated $\Phi$, $\Phi_P$, $\Phi_{TADF}$, $\Phi_{ISC}$, $\tau_{TADF}$, $k_{TADF}$, $k(T_1 \rightarrow T_2 \rightarrow S_1)$, and $k(S_1 \rightarrow T_2 \rightarrow T_1)$ values. The $\Phi$, $\Phi_P$, $\Phi_{TADF}$, and $\Phi_{ISC}$ values were quantitatively consistent with the experimental values because of the quantitative predictions for $k_F(S_1 \rightarrow S_0)$, $k_{IC}(S_1 \rightarrow S_0)$, $k_{ISC}(S_1 \rightarrow T_2)$, and $k_{ISC}(S_1 \rightarrow T_1)$. The calculated conversion rate constant $k(S_1 \rightarrow T_2 \rightarrow T_1)$ was consistent with the experimental $k_{ISC}$, suggesting that the experimental ISC should be assigned to $S_1 \rightarrow T_2 \rightarrow T_1$ ISC rather than direct $S_1 \rightarrow T_1$ ISC. All the calculated single-step rate constants and $k(S_1 \rightarrow T_2 \rightarrow T_1)$ agreed well with the experimental values. However, the agreements of the calculated $k_{TADF}$ and $k(T_1 \rightarrow T_2 \rightarrow S_1)$ ($0.16 \times 10^4 \, s^{-1}$ and $0.17 \times 10^4 \, s^{-1}$, respectively) and the experiments ($0.94 \times 10^4 \, s^{-1}$ and $1.0 \times 10^4 \, s^{-1}$, respectively) were not as good as we expected, compared with the case of $k(S_1 \rightarrow T_2 \rightarrow T_1)$. Considering that they included the uphill-energy $T_1 \rightarrow T_2$ IC, there was a possibility that the overestimation of $\Delta E(T_1 \rightarrow T_2)$ leads to an underestimation of $k_{IC}(T_1 \rightarrow T_2)$, $k_{TADF}$, and $k(T_1 \rightarrow T_2 \rightarrow S_1)$.

To examine the effect of $\Delta E(T_1 \rightarrow T_2)$ on $k_{IC}(T_1 \rightarrow T_2)$, $k_{TADF}$, and $k(T_1 \rightarrow T_2 \rightarrow S_1)$, we recalculated these rate constants using a corrected $\Delta E(T_1 \rightarrow T_2)$. The rate-determining step for the $T_2$-mediated RISC was $T_1 \rightarrow T_2$ IC; hence, $\Delta E(T_1 \rightarrow T_2)$ could be viewed as the activation energy for DABNA-1 TADF and RISC. As discussed above, the energy difference between the calculated and experimental $T_1 \rightarrow S_1$ was 60 meV. Thus, 60 meV was subtracted from the EOM-CCSD-calculated $\Delta E(T_1 \rightarrow T_2)$ value of 252 meV. The energy levels and rate constants calculated with the corrected $\Delta E(T_1 \rightarrow T_2)$ are also listed in Table 1. The value of $k(S_1 \rightarrow T_2 \rightarrow T_1)$ was unchanged, suggesting that the $T_1$ energy correction had little effect on the ISC. In contrast, $k_{IC}(T_1 \rightarrow T_2)$ increased from $1.1 \times 10^7 \, s^{-1}$ to $3.3 \times 10^8 \, s^{-1}$. As a result, $k_{TADF}$ and $k(T_1 \rightarrow T_2 \rightarrow S_1)$ increased to $1.8 \times 10^4 \, s^{-1}$ and $1.9 \times 10^4 \, s^{-1}$, respectively, which were in much better agreements with the experiments. These results suggested that the $\Delta E(T_1 \rightarrow T_2)$ overestimation was responsible for the underestimation of $k_{TADF}$ and $k(T_1 \rightarrow T_2 \rightarrow S_1)$. Thus, $k_{TADF}$ and $k(T_1 \rightarrow T_2 \rightarrow S_1)$ depended largely on $k_{IC}(T_1 \rightarrow T_2)$, which was the sum of IC rate constants for individual molecular vibrations, $k_{IC,\alpha}(T_1 \rightarrow T_2)$, where $\alpha$ denoted the $\alpha$th vibrational mode (Supplementary Equations S5–S7). A combined in-plane and out-of-plane bending mode (743 cm$^{-1}$) had the largest $k_{IC,\alpha}(T_1 \rightarrow T_2)$ value (Supplementary Table 14 and Supplementary Fig. 5), suggesting that the $T_1 \rightarrow T_2$ IC was predominantly accelerated by the bending mode.

**Summary**. In this work, we calculated the IC, ISC, RISC, fluorescence, phosphorescence, and TADF rate constants for DABNA-1, using EOM-CCSD wave functions and Fermi's golden rule. The values quantitatively reproduced those from experiments, and thus validated the calculations. Various quantum yields, including PLQY, were also predicted and we revealed the DABNA-1 decay mechanism. The calculated population kinetics and the trPL decay curve indicated that TADF from DABNA-1 occurs via consecutive two processes, $T_1 \rightarrow T_2$ IC, $T_2 \rightarrow S_1$ RISC, and $S_1 \rightarrow S_0$ radiative decay. Our proposed method here will be useful for accurate prediction of all rate constants and quantum yields relevant to OLED phenomena for various compounds with a wide range of structures.

## Methods

**Calculation of matrix elements between vibronic states.** Geometry optimization and frequency analysis of $S_0$ for DABNA-1 were performed using the TPSSh/6-31+G(d) method, whereas those of $S_1$, $T_1$, and $T_2$ were performed using the TD-TPSSh/6-31+G(d) method with spin multiplicity = 1 (Supplementary Table 1–5 and Supplementary Fig. 1). The polarizable continuum model (PCM) of a $CH_2Cl_2$ solvent was used to consider the effects on vibronic states. We used the functional and the PCM conditions for ground- and excited-state geometry optimizations because Gao et al. reported that the TPSSh/6-31+G(d)-PCM model reproduced experimental DABNA-1 emission and absorption wavelengths in $CH_2Cl_2$[41]. The geometry optimization and frequency analysis were performed with the Gaussian 16 program package[47]. For MR molecules, EOM-CCSD method and algebraic diagrammatic construction of the second order, have been shown to be suitable for calculating singlet–triplet energy differences[42,48–50]. Here, excited-state calculations were performed using the EOM-CCSD/6-31 G method implemented in the Q-Chem program package[51]. Excitation energies, SOCs, vibronic couplings, transition-dipole moments, and permanent dipole moments were calculated using EOM-CCSD wave functions. Examples of Q-Chem input files for running SOC calculations are shown in the Supplementary Methods.

**Calculation of IC rate constant.** Mathematical formulation of a method of calculating vibronic couplings and $k_{IC}$ is described elsewhere[45,52] and reviewed in Supplementary Equation S5–S9. First, for each vibrational mode $\alpha$ and each $S_m$–$S_n$ transition, $\Delta E_{FC}(S_m - S_n)$ and $V_\alpha$ were obtained from the EOM-CCSD calculations (Supplementary Equation S7) and LSF$_{IC,\alpha}$ was obtained from the frequency analyses performed using the TPSSh/6-31+G(d)-PCM model (Supplementary Equations S8 and S9). Then, $k_{IC,\alpha}(S_m - S_n)$ was calculated (Supplementary Equation S6). Finally, $k_{IC,\alpha}(S_m - S_n)$ was calculated by summing $k_{IC,\alpha}(S_m - S_n)$ over $\alpha$ (Supplementary Equation S5). This protocol was also used for calculating the IC rate constants for $T_m - T_n$ transitions.

## Data availability
All relevant data are available from the authors upon request.

## Code availability
The code used to generate Fig. 2 is available from the authors upon request.

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

## Acknowledgements

The quantum chemical calculations using the Gaussian 16 and Q-Chem program packages were performed on the SuperComputer System, Institute for Chemical Research, Kyoto University. It was also supported by JSPS KAKENHI grant numbers: 19K05629 and JP20H05840 (Grant-in-Aid for Transformative Research Areas, "Dynamic Exciton"). We thank Edanz (https://jp.edanz.com/ac) for editing a draft of this paper.

## Author contributions

K.S. performed the theoretical calculations. H.K. planned and supervised the project. All authors contributed to the writing of this paper and have approved the final version.

## Competing interests

The authors have no competing interests.

## Additional information

**Peer-review information** *Communications Chemistry* thanks Jianzhong Fan and the other, anonymous, reviewer(s) for their contribution to the peer review of this work. Peer-reviewer reports are available.

