## [Peer Review File · Communications Chemistry]

REVIEWERS' COMMENTS:

Reviewer #1 (Remarks to the Author):

The authors present a detailed study of the rISC mechanism in DABNA-1. This important molecule has been widely studied, but as stated by the authors there remains much discussion about the exact mechanism of TADF. The work quite clearly shows the importance of the T2 state and this will have important implications for the study of these molecules. I commend the authors for their excellent work and recommend it for publication.

Reviewer #2 (Remarks to the Author):

The author reported a comprehensive understanding of the multiple resonance (MR) effect-induced TADF mechanism using DABNA-1 as a model compound. Based on their rate-constant prediction method using EOM-CCSD wave functions, the k_F , k_{IC} , k_{ISC} , and k_{RISC} were calculated and in good agreement with experimental values. In addition, the calculation of population kinetics of the excited states and the transient photoluminescence (trPL) decay curve enabled the determination of the lifetime of the delayed component in trPL, the rate constant of TADF (k_{TADF}) and the PLQY, indicating that MR-TADF involved $T_1 \rightarrow T_2$ IC and $T_2 \rightarrow S_1$ RISC process. Overall, this paper is beneficial for accurate prediction of rate constants and quantum yield, leading to development of efficient MR-TADF emitter in OLED, which will be of interest to the readers of this journal. I recommend acceptance of the paper after the minor revision as listed below.

1. The journal title abbreviation of Advanced Functional Materials should be Adv. Funct. Mater. in ref 21, 28.
2. The T2 geometry is missing in the title of Supplementary Figure 2; LUMO+5, LUMO, HOMO, and HOMO-1 of DABNA-1 calculated for the S0, S1, T1 and T2 geometries.
3. p. 7 line 121: 6-31G(d) or 6-31G ? Please check all the description for the basis set.
4. It is better to provide orbital symmetry and state symmetry for each state.

Reviewer #3 (Remarks to the Author):

The paper "Comprehensive Understanding of Multiple Resonance Thermally Activated Delayed Fluorescence via Quantum Chemistry Calculations" presented by Kaji, studied the relationship between molecular structures and luminescence properties for MR-TADF molecule DABNA-1. Based on the EOM-CCSD wave functions and Fermi's golden rule, the rate constants, such as fluorescence, phosphorescence and ISC, RISC as well as the IC between T1 and T2, are all quantitatively calculated, exciton conversion processes are revealed. The work is original and should have broad appeal to physical chemists, because it elucidates the fundamental factors behind MR-TADF molecules. In terms of clarity, I understand what the authors write most of the time, I would therefore recommend publication of the paper, after addressing some points:

1. Following the equations of S5 to S9, the IC rate can be calculated. Could the author provide the calculation details and steps for DABNA-1?
2. Some abbreviations should be explained in the abstract section, such as S1 and T1.

3. What's the method for the optimization of T1, UDFT with multiplicity=3 or TD-DFT with multiplicity=1?
4. Could the author provide the script for SOC constant calculations or key words?

Our response to Reviewer 1:

We thank the reviewer for the thorough reading of our manuscript and the positive comments.

Our response to Reviewer 2:

We thank the reviewer for the thorough reading of our manuscript and the additional comments. According to the reviewer's comments, we revised the manuscript. Important changes are highlighted in yellow in the revised manuscript. Our point-to-point response to the comments is as follows.

1. The journal title abbreviation of *Advanced Functional Materials* should be *Adv. Funct. Mater.* in ref 21, 28.

According to the reviewer's comment, we corrected the journal title abbreviation of *Advanced Functional Materials* to *Adv. Funct. Mater.* in references 21 and 28.

2. The T₂ geometry is missing in the title of Supplementary Figure 2; LUMO+5, LUMO, HOMO, and HOMO-1 of DABNA-1 calculated for the S₀, S₁, T₁ and T₂ geometries.

According to the reviewer's comment, we corrected the title of Supplementary Figure 2 (page S16) from "LUMO+5, LUMO, HOMO, and HOMO-1 of DABNA-1 calculated for the S₀, S₁, and T₁ geometries" to "LUMO+5, LUMO, HOMO, and HOMO-1 of DABNA-1 calculated for the S₀, S₁, T₁, and T₂ geometries".

3. p. 7 line 121: 6-31G(d) or 6-31G ? Please check all the description for the basis set.

The basis set used for the EOM-CCSD calculations is 6-31G. We corrected the basis set description from "6-31G(d)" to "6-31G" in the revised manuscript (page 6 and Table 1).

4. It is better to provide orbital symmetry and state symmetry for each state.

The revised manuscript shows orbital symmetry and state symmetry in Figures 1a and 1b, respectively.

Our response to Reviewer 3:

We thank the reviewer for the thorough reading of our manuscript and the additional comments. According to the reviewer's comments, we revised the manuscript. Important changes are highlighted in yellow in the revised manuscript. Our point-to-point response to the comments is as follows.

1. Following the equations of S5 to S9, the IC rate can be calculated. Could the author provide the calculation details and steps for DABNA-1?

Our response:

We provided the calculation details and steps for DABNA-1 in the revised manuscript (page 11).

2. Some abbreviations should be explained in the abstract section, such as S1 and T1.

Our response:

We explained the abbreviations of S₁ and T₁ in the revised manuscript (Abstract and Introduction sections).

3. What's the method for the optimization of T1, UDFT with multiplicity=3 or TD-DFT with multiplicity=1?

Our response:

The T₁-geometry was optimized using the TD-TPSSH/6-31+G(d) method with multiplicity=1. In the revised manuscript (page 10), we clearly describe the spin multiplicity of the triplet state.

4. Could the author provide the script for SOC constant calculations or key words?

Our response:

The SOC constants were calculated using the Q-Chem program package. In the revised Supplementary Information (page S3), we described key words for running SOC calculation with the Q-Chem program package. We added a citation for the Q-Chem program package as reference 2 in the revised Supplementary Information (page S39).